# Postpartum Endometritis and Sepsis Associated with *Gardnerella vaginalis* and *Anaerococcus tetradius*: Case Report and Literature Review

**DOI:** 10.3390/reports8030143

**Published:** 2025-08-10

**Authors:** Justina Martikaitytė, Agnė Bartulevičienė, Virginija Paliulytė, Darius Dasevičius, Diana Ramašauskaitė

**Affiliations:** 1Faculty of Medicine, Vilnius University, 03101 Vilnius, Lithuania; justina.martikaityte@mf.stud.vu.lt (J.M.); virginija.paliulyte@mf.vu.lt (V.P.); diana.ramasauskaite@mf.vu.lt (D.R.); 2Vilnius University Hospital Santaros Clinics, 08406 Vilnius, Lithuania; 3National Center of Pathology, 08406 Vilnius, Lithuania; darius.dasevicius@santa.lt

**Keywords:** postpartum endometritis, abdominal wound infection, sepsis, *Gardnerella vaginalis*, *Anaerococcus tetradius*

## Abstract

**Background and Clinical Significance:** *Anaerococcus tetradius* (*A. tetradius*) and *Gardnerella vaginalis* (*G. vaginalis*) are rare etiological factors for postpartum endometritis and are typically associated with bacterial vaginosis. However, in some cases, *G. vaginalis* and *A. tetradius* can cause serious postpartum endometritis with complications such as sepsis. **Case Presentation:** 26-year-old pregnant woman expecting monochorionic diamniotic twins presented to the hospital at 35 weeks and 3 days of gestation and two male infants were delivered via the Cesarean section. On the fifth day after delivery, the patient began to complain of intense abdominal pain, a fever of 37.9 °C, and overall weakness. Blood tests revealed neutrophilic leukocytosis, increased C-reactive protein (CRP) of 225.4 mg/L. Upon examination, abdominal distension, tenderness on palpation, and positive symptoms of peritoneal irritation were present and the site of the abdominal incision was inflamed with flowing foul-smelling greenish pus. Ultrasound examination revealed free fluid collection in the peritoneal cavity, under the liver, and around the uterus. Later, the condition of the patient worsened with progressing hypotension and respiratory distress. As a result, suppurative peritonitis and sepsis was suspected and the patient underwent urgent total hysterectomy without oophorectomy. Acute endometritis, focal myometritis, and chronic cervicitis were concluded from histopathological examination of the removed uterus. Microbiological tests showed the most abundant growth of *A. tetradius* in the wound cultures and great abundance of *G. vaginalis* in the abdominal cavity cultures. After trying three different treatment schemes and difficulties with determining the antibiotic sensitivity tests for pathogens, the antibacterial therapy was escalated to Meropenem, which was found to be effective, and the patient was discharged home. **Conclusions:** This case report highlights the severity of complications of postpartum endometritis that can be caused by rare pathogens (such as *G. vaginalis* and *A. tetradius*), and strategies for how to manage it. The clinical presentation of a patient should be monitored closely for several days after Cesarean section and if endometritis is suspected, microbiological cultures are necessary to determine the cause of infection and implement an appropriate treatment.

## 1. Introduction

Postpartum endometritis is an acute condition when the endometrium of the uterus becomes inflamed after birth. The infection is typically caused by ascending cervical and vaginal pathogens [1]. The disease is more prevalent after Cesarean section and is mainly caused by *Streptococcus pyogenes* and *Staphylococcus aureus*. The other most common pathogens mentioned in the scientific literature from 1985 to 2023 that cause chronic endometritis are *Chlamydia* and *Ureaplasma* (more than 10%), followed by *Streptococcus*, *Mycoplasma*, and *Enterococcus* (approximately 10%) [2]. *Gardnerella vaginalis* (*G. vaginalis*) was mentioned in only around 4% of the literature as a potential cause of endometritis, and *Anaerococcus tetradius* (*A. tetradius*) was not mentioned at all [2]. Gardnerella vaginalis is a Gram-negative or Gram-variable facultative anaerobic rod-shaped bacterium, first identified in 1955 by H. L. Gardner. It constitutes approximately 1% of the normal vaginal flora [3]. When this starts to dominate rather than the *Lactobacillus* species, the vagina undergoes an infection called bacterial vaginosis [3,4]. The *Gardnerella* genus comprises over ten species, of which only six have been validly described [5]. Differentiation among these species relies on either 16S ribosomal RNA gene analysis, which is not entirely reliable due to gene similarities in different species, or the amplification of the *cpn60* gene using PCR [5]. *Anaerococcus tetradius* is a Gram-positive, anaerobic, non-motile coccus that produces butyrate and has saccharolytic properties. This pathogen has also been linked to bacterial vaginosis [6,7,8]. The genus was first described in 2001, and fifteen species have been reported to date [8,9]. Even though they can also be found in different infectious discharges such as foot ulcers, upper respiratory cavities, genital tract abscesses, or on the skin, the majority of the species are isolated from the human vagina [8]. Matrix-assisted laser-desorption (MALDI-TOF) MS protein analysis, along with a reference mass spectrum or the scheme of multiplex PCR, can help identify different species of *Anaerococcus* [8,10]. In rare cases, *G. vaginalis* and *A. tetradius* can cause serious postpartum endometritis with complications such as sepsis, and this clinical case report will discuss this.

## 2. Case Report

A 26-year-old woman expecting monochorionic diamniotic twins presented to our hospital at 35 weeks and 3 days of gestation with regular uterine contractions and cervical dilation of 6 cm. Delivery was planned via Cesarean section due to breech presentation of the first twin. Two male infants were born, weighing 2400 and 2850 g with Apgar scores 9 and 8 at 1 min, and 9 and 9 at 5 min, respectively. Two grams of intravenous cefazolin was injected before surgery as preoperative antibiotic prophylaxis. The placentas of the twins were sent for histopathological examination. Both newborns showed no signs of systemic infection after birth; however, the second twin was diagnosed with conjunctivitis. The patient did not have any complaints for the first four post-operative days, but on the fifth day after delivery, started complaining of acute onset abdominal pain, a fever of 37.9 °C, and general weakness. Her blood pressure (BP) was 109/65 mmHg with a pulse of 125 beats per minute (bpm) and SpO2 of 98%. Blood tests revealed neutrophilic leukocytosis, microcytic, hypochromic anemia, and increased C-reactive protein (CRP) of 225.4 mg/L. Upon examination, abdominal distension, tenderness on palpation, and positive symptoms of peritoneal irritation were present. The site of the abdominal incision was inflamed, painful, and red. About 100 mL of foul-smelling greenish pus began to flow after opening the abdominal incision, and similar vaginal discharge was also observed. Ultrasound examination revealed free fluid collection in the peritoneal cavity, under the liver, and around the uterus. Empirical antibiotic therapy of piperacillin-tazobactam (4.5 g *i*/*v*) and metronidazole (500 mg *i*/*v*) was started.

The condition of the patient worsened with progressing hypotension (systolic blood pressure < 90 mmHg), tachypnea (>25 breaths per minute), and respiratory distress. As a result, suppurative peritonitis and sepsis was suspected and the patient underwent urgent relaparotomy. The same Cesarean section incision site was used to open the abdomen. A total of 1500 mL of greenish foul-smelling pus was observed in the abdominal cavity and subsequently drained. The uterus was gray, covered in fibrin, and uterine scar dehiscence was observed. Furthermore, the sides of the uterine scar were filled with necrotic masses, and fibrin depositions were observed on the intestines and in the pouch of Douglas. Due to the advanced suppuration and necrosis, the multidisciplinary team concluded that the uterine-sparing procedure was not feasible. Therefore, a total hysterectomy without oophorectomy was performed. The removed uterus was sent for histopathological examination, and microbiological culture from the wound and abdomen was taken.

After relaparotomy and hysterectomy, the patient was further treated in the intensive care unit. Due to severe anemia (Hb 59 g/L), 5 units of erythrocyte mass were transfused. On the first post-operative day CRP of 181.3 mg/L and procalcitonin of 2.72 μg/L were observed, thus empirical antibiotic therapy of Piperacillin-tazobactam (4.5 g *i*/*v* 4 times per day) and Clindamycin (900 mg *i*/*v* 3 times per day) was continued. On the second post-operative day, preliminary microbiological test results were received, showing growth of *Anaerococcus tetradius*, *Gardnerella vaginalis*, and *Staphylococcus epidermidis* in the wound cultures and growth of *Gardnerella vaginalis* and *Fusobacterium gonidiaformans* in the cultures taken from the abdominal cavity (microorganisms were identified using MALDI-TOF mass spectrometry). On the fourth day after the surgery, more detailed microbiological test results were received, showing the most abundant growth of *Anaerococcus tetradius* in the wound cultures and great abundance of *Gardnerella vaginalis* in the abdominal cavity cultures. Antibiotic sensitivity testing revealed that the cultivated type of *Gardnerella vaginalis* was sensitive to Clyndamicin and Pyperacillin-tazobactam, but resistant to Metronidazole, thus the previously started antibioticotherapy was not changed. Unfortunately, antibiotic sensitivity for both *Anaerococcus tetradius* and *Fusobacterium gonidiaformans* could not be determined, even though the tests were repeated and carried out for a period of ten days. On the fifth day, during the wound dressing change, suppurative wound edges were observed. Upon applying pressure, secretion of pus was present. Skin sutures were removed and the dehiscence throughout the length of the wound up to the level of aponeurosis was observed. The wound was thoroughly washed with hypertonic sodium chloride and iodine solution, then packed with setons immersed in hypertonic solution. Secondary wound healing was stated, and daily wound dressing with hypertonic solution washing and seton packing was performed (see Figure 1 below). As the condition of the patient improved, she was transferred to the obstetrics unit. On the seventh day of antibacterial therapy, inflammatory markers had decreased (CRP 81.3 mg/L, procalcitonin of 0.11 μg/L), but as the secondary wound healing was observed, the patient was consulted by a clinical pharmacist, and it was decided to continue the previous treatment. On the tenth post-operative day, the patient presented with a fever of 38 °C, and a slight increase in CRP was observed (up to 93 mg/L). Cultures were taken from the intravenous catheter as well as urine, and repeated consultation with a clinical pharmacist was carried out. Since the antibiotic sensitivity tests for *A. tetradius* and *F. gonidiaformans* could not be determined, it was decided to escalate the antibacterial therapy to Meropenem (1 g *i*/*v* 3 times a day). The previously mentioned intravenous catheter and urine cultures came back negative. After five days of treatment with Meropenem, inflammatory markers dramatically decreased, and the patient was discharged home with further treatment of oral metronidazole (500 mg 3 times a day) for an additional week.

The histopathological test results were received. Placentas, placental membranes, and umbilical cords showed no signs of inflammatory infiltration (see Figure 2, Figure 3 and Figure 4 below). Acute endometritis, focal myometritis, and chronic cervicitis were concluded from histopathological examination of the removed uterus (see Figure 5 and Figure 6 below).

## 3. Discussion

*Gardnerella vaginalis*, primarily transmitted through sexual contact, is commonly associated with bacterial vaginosis. However, in rare cases, it can act as a pathogen responsible for life-threatening conditions such as endometritis [1,11]. *G. vaginalis* attaches to the vaginal epithelium and secretes specific substances, such as cytotoxin and proteolytic enzymes, which cause the death of epithelial cells and result in a specific fishy odor from the genitals—a very distinctive symptom of bacterial vaginosis [12].

However, the role of this pathogen in postpartum endometritis, which can manifest as a fever, abdominal discomfort and pain, changed vaginal discharge, or bleeding from the genitals, is still under research [1]. Only a few scientific articles examined the role of *G. vaginalis* and bacterial vaginosis in postpartum endometritis. However, all of them concluded that it is highly associated with post-cesarean section endometritis, and its incidence increased by six times compared to the control group [13,14,15]. Other microorganisms that are common in bacterial vaginosis and can lead to postpartum endometritis include *Peptococcus* spp., *Bacteroides* spp., *Staphylococcus epidermidis*, *Streptococcus agalactiae*, and *Ureaplasma urealyticum* [15].

Bacterial vaginosis is also extremely relevant in other obstetrical complications, such as chorioamnionitis, postpartum fever, and preterm birth [15]. All of these secondary infections due to bacterial vaginosis require extensive treatment, such as increased dosage or variety of antibiotics, prolonged hospitalization time, and a high chance surgical intervention. Bacterial vaginosis is diagnosed in up to twenty percent of patients during pregnancy [15].

Postpartum endometritis is the most common obstetrical infection, especially after delivering a fetus by cesarean section [16]. It is mainly polymicrobial—caused by aerobic and anaerobic bacteria. Endometritis is more likely to occur in an already damaged endometrium, especially during the interventions of delivery. Other risk factors include chorioamnionitis, maternal infections such as HIV and obesity, manual removal of the placenta, and prolonged rupture of membranes. In a biopsy, the infiltration of neutrophils in the endometrium is observed, physical examination is marked by uterine tenderness and its increased size, and leukocytosis may be present. In some cases, postpartum endometritis can be complicated by abscesses, sepsis, necrotizing myometritis or fasciitis, and peritonitis, which may require urgent surgery [17,18].

Even though there is little research on the role of *G. vaginalis* in postpartum endometritis and its complications, there is some evidence that several microorganisms causing bacterial vaginosis are linked to septic shock. Even though *G. vaginalis* bacteremia is especially rare, there are some clinical case reports where it is discussed. Research by P. Taillandier et al. revealed that *G. vaginalis* and *Atobopium vaginae* were responsible for the development of peritonitis and later septic shock after a hysterectomy and adnexectomy [19]. Another study reported the risk factors of septic shock caused by *G. vaginalis*, such as binge drinking, metabolic diseases like diabetes, and immunosuppression. However, our patient did not have any pre-existing conditions, but serious complications were still present [20].

The treatment guidelines for septic shock related to *G. vaginalis* are very limited, as this complication is especially uncommon. A similar case report after a cesarean section was described 2 years ago in China, and specific management was found to be effective [21]. Primarily, after detecting *G. vaginalis* in the blood, treatment with cefoperazone-sulbactam (3 g) was started. However, after 5 days of treatment, inflammation markers in the blood were still high, so ornidazole (0.5 g) was added, and this therapy was continued for 10 days, resulting in full patient recovery. This extended combination of different antibiotics might be needed due to increased resistance to treatment of *G. vaginalis*, because of biofilm formation in the organism [18]. Other scientific data were used to analyze treatment options for *G. vaginalis*-linked septic shock. It was found that ceftriaxone, ornidazole, and levofloxacin therapy was effective, as was a combination of penicillin and metronidazole or cefoperazone-sulbactam and metronidazole, or even cefuroxime alone [21,22]. Another study found that a combination of oral antibiotics—ampicillin and clindamycin—was also successful in treating *G. vaginalis* bacteremia [23]. Complexity of antibacterial treatment was also seen in our presented case—three different treatment schemes were tried until successful effect was reached.

*A. tetradius* is a part of the normal vaginal microbiota and its relationship with postpartum endometritis is unclear, as to this day, to our knowledge, there is no literature published on this topic. However, the role of *A. tetradius* in bacteremia is being examined, and genital surgery is considered a risk factor for anaerobe infection [24]. In another study, it was found that *G. vaginalis* and *A. tetradius*, as a part of the vaginal microbiota, can predispose patients to human papillomavirus and its higher severity [25]. Further studies are required to investigate the role of this pathogen, since, as described in our case, it can play a crucial role in the development of postpartum endometritis and its complications.

## 4. Conclusions

Our case report highlights how an imbalance in normal vaginal microbiota sometimes leads to serious postpartum endometritis and sepsis after delivery, and the importance of a timely diagnosis. The clinical presentation of a patient should be followed for several days after Cesarean section, and when indications occur, examination of the patient is vital. Relaparotomy and microbiological culture are essential when secondary peritonitis is suspected to determine the cause of infection and to apply an appropriate treatment to avoid further complications. Even the rarest pathogens, such as *G. vaginalis* and *A. tetradius*, can cause post-cesarean endometritis, which can later on manifest as sepsis.

## Figures and Tables

**Figure 1 reports-08-00143-f001:**
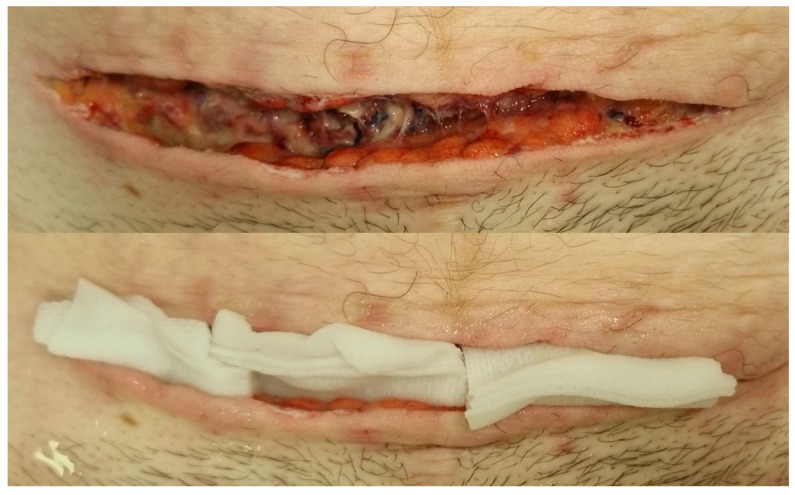
Secondary wound healing and wound packing with setons immersed in hypertonic sodium chloride solution.

**Figure 2 reports-08-00143-f002:**
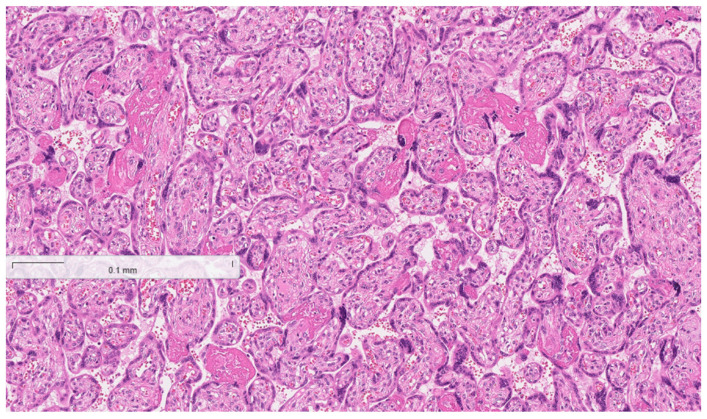
Normal histological structure of placenta (no signs of inflammatory infiltration).

**Figure 3 reports-08-00143-f003:**
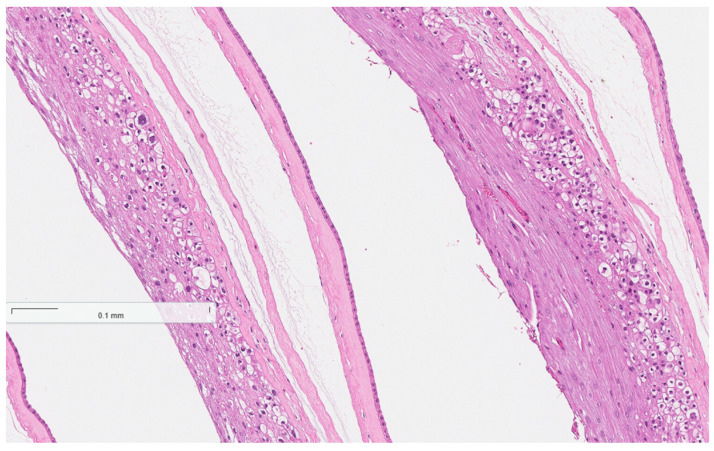
Normal histological structure of placental membranes (no signs of inflammatory infiltration).

**Figure 4 reports-08-00143-f004:**
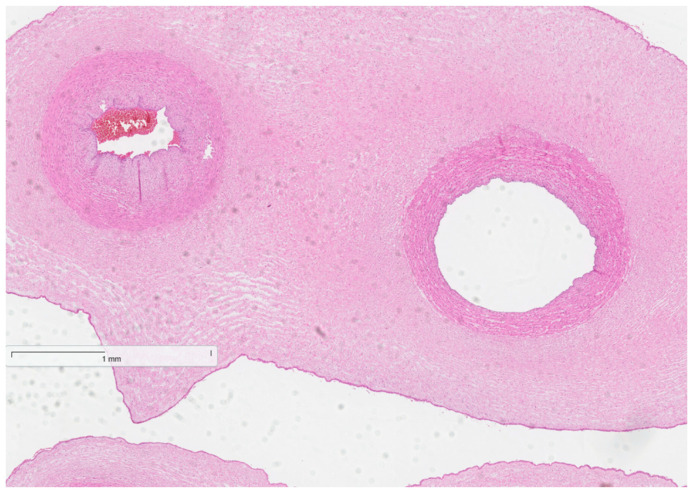
Normal histological structure of the umbilical cord (no signs of inflammatory infiltration).

**Figure 5 reports-08-00143-f005:**
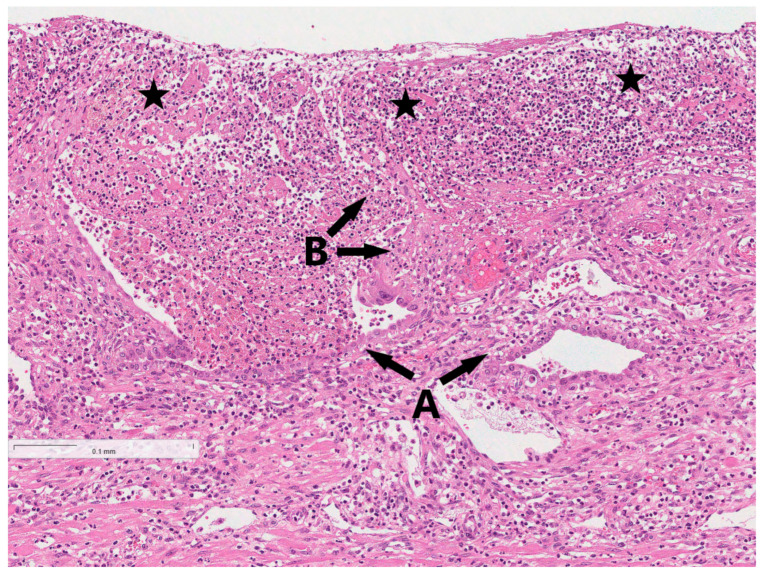
Endometrium with sparse endometrial glands (A) and moderate to dense intraglandular infiltration of neutrophilic lymphocytes (B), with partial or complete destruction of endometrial glands (asterisks).

**Figure 6 reports-08-00143-f006:**
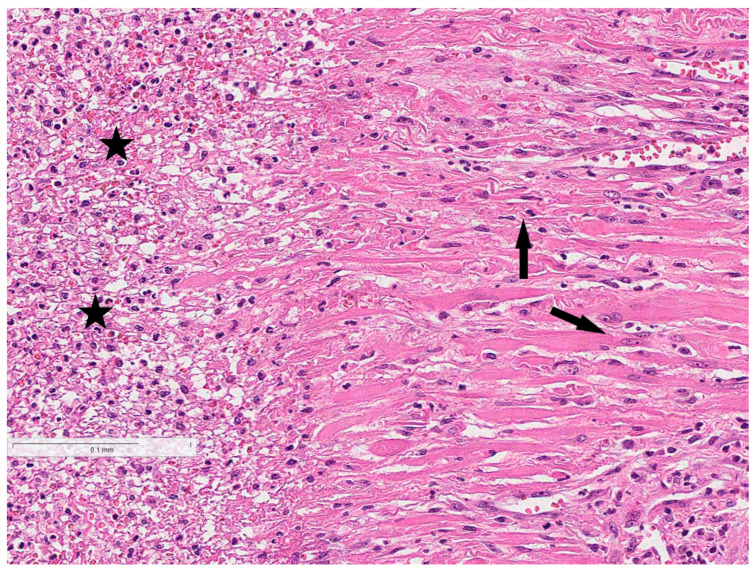
Dense infiltration of neutrophilic lymphocytes with cellular debris (asterisk) in the myometrium, with retained smooth muscle cells on the right side (arrows).

## Data Availability

The original contributions presented in this study are included in the article. Further inquiries can be directed to the corresponding author.

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
