# Peer review of "Postpartum Endometritis and Sepsis Associated with Gardnerella vaginalis and Anaerococcus tetradius: Case Report and Literature Review"

_reports, 2025, doi:10.3390/reports8030143_

Round 1

Reviewer 1 Report

Comments and Suggestions for Authors

Dear Authors,
I read your case report with great interest.
In my opinion, to make the article more accessible to readers, it would be helpful to highlight the differences between G. Vaginalis and A. Tetradius endometritis and other types of endometritis. My suggestions for improving the article are:
- If applicable, describe the risk factors for G. vaginalis and A. Tetradius endometritis.
- Describe the greater or lesser risk of postpartum endometritis compared to data in the literature compared to other pathogens.
- Describe the greater risk of destructive surgical treatment if the endometritis is caused by G. vaginalis and A. Tetradius or other pathogens.
- Describe the recommendations for first-line medical therapy if the endometritis is caused by G. vaginalis and A. Tetradius

Best Regards

Comments on the Quality of English Language

The manuscript needs a revision of language by native English Speaker

Author Response

Comments 1: ‘If applicable, describe the risk factors for G. vaginalis and A. Tetradius endometritis.’

Response 1: The general risk factors of endometritis have already been discussed in the ‘Discussion’ section, pages 5, 3rd paragraph, lines 3-7, which include many obstetrical complications, the mother’s preexisting infections, or other infections during the pregnancy. The differences in risk factors for G. vaginalis and A. tetradius endometritis were not highlighted in the scientific literature, except that bacterial vaginosis is mainly a risk factor for G. vaginalis endometritis after cesarean delivery, and A. tetradius is not typically involved (page 5, 1st paragraph, lines 3-6). Furthermore, as it is already written in our manuscript, there is little research about the role of G. vaginalis in postpartum endometritis, and even less scientific literature about A. tetradius endometritis, but the information we found is written in the ‘Discussion’ section (the risk factors of septic shock and bacteremia of G. vaginalis and A. tetradius) - page 5, 4th paragraph, lines 3-10 and page 6, 1st paragraph, lines 1-4.

Comments 2: ‘Describe the greater or lesser risk of postpartum endometritis compared to data in the literature compared to other pathogens.’

Response 2: Thank you for pointing this out. As it is written in the ‘Introduction’ section, postpartum endometritis is mainly caused by Streptococcus pyogenes and Staphylococcus aureus pathogens (page 1, 1st paragraph, lines 3-4). The more detailed statistics were reviewed, and we found an article (doi: 10.5935/1518-0557.20240088) investigating the principal bacteria associated with chronic endometritis. The main pathogens mentioned in the scientific literature from 1985 to 2023 that cause chronic endometritis are Chlamydia and Ureaplasma (more than 10%), followed by Streptococcus, Mycoplasma, and Enterococcus (around 10%). G. vaginalis was mentioned in only around 4% of the literature as a potential cause of endometritis, and A. tetradius was not mentioned at all. We have provided this additional information to our manuscript, it can be found in the ‘Introduction’ section, page 1, 1st paragraph, lines 4-9.

Comments 3: ‘Describe the greater risk of destructive surgical treatment if the endometritis is caused by G. vaginalis and A. Tetradius or other pathogens.’

Response 3:  Thank you for this suggestion. It would have been interesting to explore this aspect. However, as there is no literature about A. tetradius endometritis, and not a lot of information about G. vaginalis endometritis, it is not possible to describe it. After doing another revision of the scientific literature about this topic, it was found that only strategies of conservative management of postpartum endometritis were analyzed, and the frequency of surgical treatments was not explored. It was only mentioned that complications of postpartum endometritis, such as peritonitis, pelvic abscess, septic shock, or septic pelvic thrombophlebitis, should be treated with surgical interventions, and the most destructive being hysterectomy (doi: 10.5935/1518-0557.20220015). The possible risk of G. vaginalis and A. tetradius ending in destructive surgical management more frequently than endometritis caused by other pathogens is an extremely interesting topic, which should be analyzed further.

Comments 4: ‘Describe the recommendations for first-line medical therapy if the endometritis is caused by G. vaginalis and A. Tetradius’

Response 4: We agree that this point is very important and should be described. The treatment of septic shock and endometritis caused by G. vaginalis and A. tetradius, which was found to be effective in other case reports and our study, is described in the ‘Discussion’ section (page 5, 5th paragraph). Other scientific literature mainly suggests using broad-spectrum antibiotics, and the first-line medical therapy for endometritis is considered to be a combination of clindamycin and an aminoglycoside like gentamicin (doi: 10.1002/14651858.CD001067.pub3). No other first-line recommendations for treating endometritis caused by G. vaginalis and A. tetradius were found.

Comments on the Quality of English Language (The manuscript needs a revision of language by native English Speaker).

Response: The manuscript has been reviewed and revised by a native English speaker.

Reviewer 2 Report

Comments and Suggestions for Authors

Dear authors, it is an interesting topic. More information regarding such rare infection during postpartum are necessary as background.

In the case report you described all the relevant details regarding patient s evolution. I would like to know how was the evolution of the baby s. Were any signs of infection observed in newborns?

You said you sent the placenta for histological examination. It would be also relevant to know if there were any signs of infection, slides would be useful.

Also images of the uterus if you have would be useful, as you have with the post-cesarean wound.

Otherwise, the case presentation is interesting and deserves to be published.

Author Response

Comments 1: In the case report you described all the relevant details regarding patient’s evolution. I would like to know how was the evolution of the baby’s. Were any signs of infection observed in newborns

Response 1: Thank you for requesting this information. The first twin did not show any signs of infection while the second twin was diagnosed with conjunctivitis. This additional information has been added to the manuscript (page 2, paragraph 2, lines 7-9).  

Comments 2: You said you sent the placenta for histological examination. It would be also relevant to know if there were any signs of infection, slides would be useful.

Response 2: As it was already mentioned in our manuscript on page 3, paragraph 2, lines 1-2 “Placentas, placental membranes and umbilical cords showed no signs of inflammatory infiltration.” According to your comment, we have provided additional histopathological images on page 4 as figure 2, 3, and 4.

Comment 3: Also images of the uterus if you have would be useful, as you have with the post-cesarean wound.

Response 3: We completely agree with your remark. Unfortunately, as the hysterectomy was performed quite urgently, no pictures were taken during the surgery and our pathology department does not keep macroscopic images in their archive. Therefore, we have no means to provide the images of the uterus.

Reviewer 3 Report

Comments and Suggestions for Authors

Journal Reports (ISSN 2571-841X)

Manuscript ID reports-3751512

Type Case Report

Title

Postpartum Endometritis and Sepsis Associated with Gardnerella vaginalis and Anaerococcus tetradius. Case Report and Literature Review

Dear authors,

I have reviewed your manuscript concerning postpartum endometritis and sepsis associated with Gardenella vaginalis and Anaerococcuc tretadius. Postpartum infections, especially endometritis and sepsis, are interesting topic, as those complications may seriously influence postpartum period.

In your manuscript you have presented a case of postpartum endometritis and sepsis after Cesarean section, and described a case in detail concerning timing of events, laboratory findings and antibiotic treatment which was prolonged. However, you haven`t linked microbiological findings with clinical course and treatment. In order to justify your case, you should reconsider your manuscript and remodel it.

Best regards

Author Response

Comments 1: In your manuscript you have presented a case of postpartum endometritis and sepsis after Cesarean section, and described a case in detail concerning timing of events, laboratory findings and antibiotic treatment which was prolonged. However, you haven`t linked microbiological findings with clinical course and treatment. In order to justify your case, you should reconsider your manuscript and remodel it.

Response 1: Thank you for your comment and pointing out this inconvenience. As a result, we remodeled our manuscript, the first paragraph on page 4 was edited and more details of the microbiological findings were provided on page 2, paragraph 3, lines 5-9 and page 3, paragraph 1, lines 1-9.

Reviewer 4 Report

Comments and Suggestions for Authors

However, there is insufficient microbiological evidence to indicate that G. vaginalis and A. tetradius were the infectious causes. That is, there is no detailed description of how they were isolated and identified. Therefore, it is necessary to mention the microbiological details of all the isolated and identified microorganisms mentioned in the clinical case.

The introduction section lacks a detailed description of the genus Anaerococcus, as is the case with the Gardnerella genus. Furthermore, both genera have several species. For example, how did they identify that it was A. tetradius and not A. vaginalis, which is also part of the vaginal microbiota? (doi: 10.4056/sigs.2716452) Similarly, how did you identify G. vaginalis? Could it be G. piotii? (doi: 10.1093/infdis/jiae026).

The clinical case report section is adequate. However, although clinical findings are concerning and could be indicative of a severe infection, the diagnosis of sepsis typically requires evidence of organ dysfunction related to the infection (data that is not observed in the manuscript). Tachycardia is a concerning sign, but blood pressure and oxygen saturation are within normal ranges.

The discussion section is adequate. However, no evidence of isolation of G. vaginalis or A. tetradius is demonstrated.

The references section. It would be if you made a good review, some of which are indicated in the manuscript.

Comments on the Quality of English Language

Although the manuscript is very understandable, the English could be improved to convey the research more clearly.

Author Response

Comment 1: However, there is insufficient microbiological evidence to indicate that G. vaginalis and A. tetradius were the infectious causes. That is, there is no detailed description of how they were isolated and identified. Therefore, it is necessary to mention the microbiological details of all the isolated and identified microorganisms mentioned in the clinical case.
Response 1: Thank you for pointing this out. We have contacted our microbiological laboratory and they have informed us these particular pathogens have been isolated and identified using MALDI-TOF mass spectrometry. As a result, we have also provided more information about microbiological findings and added this information to our manuscript, it can be found on page 2-3, paragraph 4, lines 5-18.

Comment 2: The introduction section lacks a detailed description of the genus Anaerococcus, as is the case with the Gardnerella genus. Furthermore, both genera have several species. For example, how did they identify that it was A. tetradius and not A. vaginalis, which is also part of the vaginal microbiota? (doi: 10.4056/sigs.2716452) Similarly, how did you identify G. vaginalis? Could it be G. piotii? (doi: 10.1093/infdis/jiae026).
Response 2: Thank you for your comment. We have provided a more detailed description of the pathogens in introduction section. It can be found in revised manuscript, page 1-2, introduction section, lines 13-24. I hope the second part of the comment has been answered in the first response.

Comment 3: The clinical case report section is adequate. However, although clinical findings are concerning and could be indicative of a severe infection, the diagnosis of sepsis typically requires evidence of organ dysfunction related to the infection (data that is not observed in the manuscript). Tachycardia is a concerning sign, but blood pressure and oxygen saturation are within normal ranges.
Response 3: Thank you for this very important comment. We have noticed, that tachypnea and signs of the respiratory distress were not mentioned in the primary manuscript. The latter information can be found in revised manuscript page 2, paragraph 2, lines 1-4 . According to our valid national “Postpartum infection” guidelines, fever, leukocytosis, hypotension and tachypnea is sufficient evidence to diagnose postpartum sepsis.

Comment 4: The discussion section is adequate. However, no evidence of isolation of G. vaginalis or A. tetradius is demonstrated.
Response 5: Evidence about isolation has been provided in response 1.

Comment 5: The references section. It would be if you made a good review, some of which are indicated in the manuscript.
Response 5: The references section has been reviewed and revised.

Response to Comments on the Quality of English Language (Although the manuscript is very understandable, the English could be improved to convey the research more clearly).
Response: The manuscript has been reviewed and revised by a native English speaker.

Round 2

Reviewer 1 Report

Comments and Suggestions for Authors

Dear Authors,

congratulations for your work.

The manuscript has improved, in my opinion is suitable for the publication.

Best Regards

Reviewer 2 Report

Comments and Suggestions for Authors

Dear authors, the manuscript looks better.

Reviewer 3 Report

Comments and Suggestions for Authors Journal: Reports (ISSN 2571-841X) Manuscript ID: reports-3751512 Type: Case Report Title: Postpartum Endometritis and Sepsis Associated with Gardnerella vaginalis and Anaerococcus tetradius. Case Report and Literature Review Dear authors, thank you for the revision of your manuscript.  After requested revision, manuscript is well suited, and prepared for publication. I consider your manuscript worth publishing. Best regards

Reviewer 4 Report

Comments and Suggestions for Authors

Better information in the manuscript was observed; however, it would be helpful to comment on whether some bacteria caused the second twin's conjunctivitis (in the discussion section), as one might suspect that Chlamydia trachomatis also contributed and is present but not identified.

I do not understand why the publication date is included in some bibliographic references.
